# Dynamic Pruning of a Neural Network
# via Gradient Signal-to-Noise Ratio

**Julien Siems**[*]                                                                SIEMSJ@CS.UNI-FREIBURG.DE
*University of Freiburg*

**Aaron Klein**                                                                     KLEIAARO@AMAZON.DE
*Amazon Web Services*

**Cedric Archambeau**                                                              CEDRICA@AMAZON.COM
*Amazon Web Services*

**Maren Mahsereci**[*]                                                  MAREN.MAHSERECI@UNI-TUEBINGEN.DE
*University of Tübingen*

## Abstract

While training highly overparameterized neural networks is common practice in deep learning, research into post-hoc weight-pruning suggests that more than 90% of parameters can be removed without loss in predictive performance. To save resources, zero-shot and one-shot pruning attempt to find such a sparse representation at initialization or at an early stage of training. Though efficient, there is no justification, why the sparsity structure should not change during training. *Dynamic sparsity* pruning undoes this limitation and allows to adapt the structure of the sparse neural network during training. In this work we propose to use the gradient noise to make pruning decisions. The procedure enables us to automatically adjust the sparsity during training without imposing a hand-designed sparsity schedule, while at the same time being able to recover from previous pruning decisions by unpruning connections as necessary. We evaluate our method on image and tabular datasets and demonstrate that we reach similar performance as the dense model from which the sparse network is extracted, while exposing less hyperparameters than other dynamic sparsity methods.

## 1. Introduction

Deep neural networks have been applied successfully to several machine learning tasks spanning, for instance, computer vision and natural language processing. However, neural networks tend to reach the best performance if they are highly overparameterized (Nakkiran et al., 2020), resulting in high inference and training cost. Post-hoc pruning and retraining can often prune parameters in a network to 1/10 or 1/100 of the total number of parameters of the unpruned network and result in a better predictive performance (Han et al., 2015), raising questions of how one may prune parameters already during training. To that end, several approaches have emerged, that can be distinguished roughly by when the network is being pruned: at or close to initialization (Frankle and Carbin, 2018; Wang et al., 2019, SNIP, GASP respectively), during training (Lin et al., 2019, DPF) and post-hoc after training (Han et al., 2015). A fourth category would be iterative magnitude pruning to discover lottery tickets (Frankle and Carbin, 2018). Pruning at initialization via gradient-based weight pruning such as SNIP or GRASP find sparse networks which one can train efficiently from scratch. However, they only prune once early on and do not adapt the sparse network during training. An extensive comparison of different pruning at initialization method was done by Frankle et al. (2021) who discuss benefits and shortcomings of each method. They found that magnitude pruning after training outperformed GRASP and SNIP and that methods for pruning at initialization appear to actually suffer from the initialization and work better when used after training for a few

---

[*]. Work done while at Amazon Web Services.

iterations. Finally, they found that the layer ratios found via SNIP/GRASP are more informative than the individual weights per layer found. This work questions the utility of the pruning at initialization setting.

In this paper, we propose a method that extends Dynamic Pruning with Feedback (DPF) (Lin et al., 2019). DPF prunes during training and finds the sparse network and weights simultaneously, due to which this setting is sometimes coined 'dynamic sparsity' (Mocanu et al., 2018; Mostafa and Wang, 2019; Lin et al., 2019). A dynamic sparsity/pruning algorithm caters to the dynamics of the optimization process rather than observing and pruning at one snapshot in time. Our extension provides a novel pruning criterion which collects statistics from the optimization of the network to decide when to consider a weight for pruning and changes based on the dataset and the model architecture used. Concretely our contributions are the following:

- We propose to use the signal-to-noise ratio (SNR) of the mini-batch gradient to measure when a weight can be considered for pruning. This automatically provides a custom sparsity schedule that depends on the present dataset and network architecture.

- Using our novel sparsity criterion, we allow the number of non-zero entries in the sparsity masks to decrease and increase during training.

- We demonstrate the effectiveness of our approach on image and tabular datasets where we obtain results on par with dense learning and DPF, while gaining additional insights into the learning procedure through the sparsity schedule. In contrast to DPF, our algorithm needs no manual sparsity schedule which is a difficult parameter to set in practice and depends on the network architecture and training dataset.

## 2. Related Work

**Pruning before training/at initialization.** Several methods exist to prune at initialization. One was proposed with Lottery Ticket Hypothesis (LTH) (Frankle and Carbin, 2018; Frankle et al., 2019a). They showed that any sufficiently overparameterized network contains sparse subnetworks which train at least as quickly and generalize as well as the original unpruned network. However, they are expensive to find. Alternatively, the gradients at initialization can be used to score connections in a network and identify dominant sparse networks, e.g., via SNIP (Lee et al., 2018) or GRASP (Wang et al., 2019). In those settings the sparsity mask remains fixed during training.

**Pruning during training.** Mocanu et al. (2018) proposed sparse evolutionary training (SET) which prunes and regrows connections in a network at the end of each training epoch, following a predefined sparsity schedule. This setting was extended by Mostafa and Wang (2019) and Dettmers and Zettlemoyer (2019) by regrowing via loss gradient and by momentum magnitude respectively, allowing to remove the hand-designed sparsity. Our work is closely related to Dynamic Pruning with Feedback (DPF) (Lin et al., 2019), which is explained in more detail in Section 4. Sparsity can also be learned via gradients during training as demonstrated by Louizos et al. (2018) who uses L0 regularization, or more recently by Kusupati et al. (2020) who learn sparsity thresholds.

**Pruning after training.** This setting is the most commonly used in practice and has seen research interest for at least three decades. Prominent early examples were Optimal Brain Damage (Le Cun et al., 1989) and Optimal Brain Surgeon (Hassibi and Stork, 1992) which used approximations to the Hessian to prune neural networks. More recent approaches in this setting include Molchanov et al. (2019) and Theis et al. (2018) who prune based on the first and second order taylor expansions of the pruning problem respectively.

**Neural architecture search.** The search for a sparse neural network may also be seen as a form of neural architecture search (NAS), which can be carried out in an existing overparameterized

---

**Algorithm 1** *The detailed training procedure of pruning guided by gradient SNR.*

---

**Require:** uncompressed model weights $\boldsymbol{\theta} \in \mathbb{R}^d$, pruned weights: $\hat{\boldsymbol{\theta}}$, mask: $\mathbf{m}_{prune} \in \{0,1\}^d$; SNR exp. avg.: $\gamma$,
   burn-in steps: $bin_{steps}$, mask: $\mathbf{m} \in \{0,1\}^d$; training iterations: $T$.
1: **for** $t = 0, \dots, T$ **do**
2:     **if** $t > bin_{steps}$ **then**               ▷ trigger mask update, by default after $bin_{steps} = 1$ epoch
3:        compute SNR mask $\mathbf{m}_{snr} \leftarrow \{\mathrm{snr}(\theta_t^i) > 1 \mid \text{i in } |\theta_\mathbf{t}|\}$    ▷ let $sp_{snr}$ be the sparsity of the resulting mask
4:        compute MAG mask $\mathbf{m}_{mag} \leftarrow \{|\theta_t^i| > \gamma \mid \text{i in } |\theta_\mathbf{t}|\}$            ▷ $\gamma$ cut-off weight mag. acc to $sp_{snr}$
5:        compute PRUNE mask $\mathbf{m}_{prune} \leftarrow \mathbf{m}_{mag} \wedge \mathbf{m}_{snr}$       ▷ only prune, if $\mathbf{m}_{mag}$ and $\mathbf{m}_{snr}$ agree to prune
6:     **end if**
7:     $\hat{\boldsymbol{\theta}}_t \leftarrow \mathbf{m}_{prune} \odot \boldsymbol{\theta}_t$                                          ▷ apply resulting mask
8:     compute (mini-batch) gradient $\nabla L_{\mathcal{B}}(\hat{\boldsymbol{\theta}})$            ▷ forward/backward pass with pruned weights $\hat{\boldsymbol{\theta}}_t$
9:     update $\mathbf{m}_{snr,t}$                                  ▷ update SNR exp mov. avg. per weight
10:    $\boldsymbol{\theta}_{t+1} \leftarrow$ gradient update $\nabla L_{\mathcal{B}}(\hat{\boldsymbol{\theta}})$ to $\boldsymbol{\theta}_t$        ▷ via arbitrary optimizer (e.g. SGD with momentum)
11: **end for**
**Ensure:** $\boldsymbol{\theta}_T$ and $\hat{\boldsymbol{\theta}}_T$

---

network via dense to sparse pruning, similar to one-shot NAS (Liu et al., 2018), but may also include regrowing new connections (Mocanu et al., 2018; Dettmers and Zettlemoyer, 2019) like in evolutionary NAS (Real et al., 2019; Elsken et al., 2018). Sparse networks have also seen usage in continual learning (Cheung et al., 2019; Wortsman et al., 2020) where sparse subnetworks of a large neural network are used to avoid catastrophic forgetting.

## 3. Background: Signal-To-Noise Ratio of a Mini-Batch Gradient

This section introduces the signal-to-noise ratio (SNR) of a mini-batch gradient; Section 4 shows how it will be used in the dynamic pruning algorithm. The gradient SNR assesses the reliability of a mini-batch gradient with respect to the noise resulting from the individual datapoints in the mini-batch at every training step. Let $\mathcal{B} = \{(\mathbf{x}_i, y_i)\}_{i=1}^{|\mathcal{B}|}$ be a mini-batch with $(\mathbf{x}_i, y_i) \sim p(\mathbf{x}, y)$ i.i.d. draws and let $\nabla \mathcal{L}(\boldsymbol{\theta})$ be the true but unknown gradient of the risk $\mathcal{L}(\boldsymbol{\theta})$ with respect to the weights $\boldsymbol{\theta}$ of the neural network $f_{\boldsymbol{\theta}}(\mathbf{x})$. Furthermore, let $\Sigma_{\mathcal{B}}(\boldsymbol{\theta}) := {}^{\mathrm{Cov}[\nabla \ell_i(\boldsymbol{\theta})]}/_{|\mathcal{B}|}$ be the covariance of the mini-batch gradient $\nabla L_{\mathcal{B}}(\boldsymbol{\theta}) = \sum_i {}^{\nabla \ell_i(\boldsymbol{\theta}))}/_{|\mathcal{B}|}$ with gradients for individual datapoints denoted by $\nabla \ell_i(\boldsymbol{\theta}) := \nabla \ell(f_{\boldsymbol{\theta}}(\mathbf{x}_i), y_i)$. Due to the i.i.d assumption on the training data one can motivate the normal distribution $\nabla L_{\mathcal{B}}(\boldsymbol{\theta}) \sim \mathcal{N}(\nabla \mathcal{L}(\boldsymbol{\theta}), \Sigma_{\mathcal{B}}(\boldsymbol{\theta}))$ via the central-limit theorem.

Mahsereci et al. (2017) use this distribution $\nabla L_{\mathcal{B}} \sim \mathcal{N}$ to construct the following criterion: If the statistic $z(\boldsymbol{\theta}) := \log p(\nabla L_{\mathcal{B}}(\boldsymbol{\theta}) | \nabla \mathcal{L}(\boldsymbol{\theta}) = 0)$ coincides with its expectation $\mathbb{E}_{\mathbf{x}}[z(\boldsymbol{\theta})]$, then the mini-batch gradient is not informative anymore as it can be fully explained by sample noise and a vanishing, underlying true gradient $\nabla \mathcal{L}(\boldsymbol{\theta}) = 0$. More formally, per weight $\boldsymbol{\theta}_k$, and with the simplification $\Sigma_{\mathcal{B}}(\boldsymbol{\theta}) = {}^{\mathrm{diag}(\mathrm{Var}[\nabla \ell_i(\boldsymbol{\theta})])}/_{|\mathcal{B}|}$, it happens when $snr(\boldsymbol{\theta}_k) \leq 1$, where $snr(\boldsymbol{\theta}_k) := {}^{\nabla L_{\mathcal{B},k}^2(\boldsymbol{\theta})}/_{\Sigma_{\mathcal{B},kk}(\boldsymbol{\theta})}$ is the signal-to-noise ratio (SNR). In practice, the gradient variances $\Sigma_{\mathcal{B},kk}(\boldsymbol{\theta})$ required for the SNR can be efficiently estimated during backpropagation from the mini-batch itself (Dangel et al., 2020).

## 4. Method

Our work builds upon Dynamic Pruning with Feedback (DPF) (Lin et al., 2019) which performs weight magnitude pruning according to a predefined pruning schedule during training. Crucially, the method allows to recover from sub-optimal pruning decisions early on during training by updating the parameters of the underlying unpruned model alongside the pruned model and periodically recomputing the weight magnitude mask based on the unpruned weights, allowing weights to reenter the pruned network. Due to the simplicity of this method it is a good baseline for the dynamic pruning setting, which the authors demonstrated to work well in practice.

However, two drawbacks of DPF should be noted. First, the sparsity schedule has to be manually set prior to training and is the same regardless of the dataset and model. Second, we argue that

Table 1: Top-1 test accuracy of different baseline methods for different target sparsity ratios. We record the final achieved sparsity ratio for our method in brackets.

| Dataset | Baseline on dense model | Methods | | | | Target Pr. Ratio |
| | | Ours | Lottery Ticket | GRASP - ABS | DPF | |
|---|---|---|---|---|---|---|
| Electricity | $0.154 \pm 0.721e\text{-}3$ | $0.157 \pm 2.095e\text{-}3$ (Sp. $0.64 \pm 0.02$) | $0.2 \pm 1.873e\text{-}3$ $0.190 \pm 2.725e\text{-}3$ $0.429 \pm 0.0$ | $0.153 \pm 0.822e\text{-}3$ $0.156 \pm 1.066e\text{-}3$ $0.162 \pm 2.174e\text{-}3$ | $0.153 \pm 0.904e\text{-}3$ $0.153 \pm 1.4e\text{-}3$ $0.154 \pm 1.549e\text{-}3$ | 0.7 0.8 0.95 |
| Covertype | $0.158 \pm 1.218e\text{-}3$ | $0.16 \pm 1.322e\text{-}3$ (Sp. $0.8 \pm 0.02$) | $0.253 \pm 5.938e\text{-}3$ $0.227 \pm 0.436e\text{-}3$ $0.533 \pm 0.0$ | $0.158 \pm 2.567e\text{-}3$ $0.158 \pm 2.431e\text{-}3$ $0.148 \pm 2.937e\text{-}3$ | $0.155 \pm 2.701e\text{-}3$ $0.153 \pm 1.5e\text{-}3$ $0.156 \pm 1.366e\text{-}3$ | 0.7 0.8 0.95 |
| FMNIST | $0.0863 \pm 2.834e\text{-}3$ | $0.0934 \pm 2.593e\text{-}3$ (Sp. $0.76 \pm 0.01$) | $0.0989 \pm 1.597e\text{-}3$ $0.1008 \pm 2.024e\text{-}3$ $0.9 \pm 0.0$ | $0.0953 \pm 2.317e\text{-}3$ $0.0983 \pm 2.428e\text{-}3$ $0.104 \pm 2.310e\text{-}3$ | $0.0915 \pm 2.522e\text{-}3$ $0.0945 \pm 3.827e\text{-}3$ $0.1049 \pm 2.276e\text{-}3$ | 0.7 0.8 0.95 |
| MNIST | $8.233 \pm 0.17$ | $9.4e\text{-}3 \pm 0.668e\text{-}3$ (Sp. $0.71 \pm 0.18$) | $12.1e\text{-}3 \pm 0.294e\text{-}3$ $12.13e\text{-}3 \pm 0.047e\text{-}3$ $0.8865 \pm 0.0$ | $8.8e\text{-}3 \pm 0.51e\text{-}3$ $10.047e\text{-}3 \pm 0.826e\text{-}3$ $16.033e\text{-}3 \pm 2.38e\text{-}3$ | $9.033e\text{-}3 \pm 1.053e\text{-}3$ $9.733e\text{-}3 \pm 0.818e\text{-}3$ $11.6e\text{-}3 \pm 0.082e\text{-}3$ | 0.7 0.8 0.95 |
| CIFAR-10 | $0.076 \pm 1.975e\text{-}3$ | $0.083 \pm 1.686e\text{-}3$ (Sp. $0.81 \pm 0.03$) | - | - | $0.0892 \pm 1.37e\text{-}3$ $0.0894 \pm 1.476e\text{-}3$ $0.096 \pm 2.217e\text{-}3$ | 0.7 0.8 0.95 |

dynamic pruning algorithms should consider the gradients during training to make pruning decisions, as these are more meaningful at the beginning of the optimization than only the weight magnitude due to the random weight initialization. In addition to these drawbacks the number of steps between mask updates (16 SGD steps) was found using hyperparameter optimization, leading to defaults which may not transfer to other settings and defeating the purpose of pruning.

We propose to use the mini-batch gradient SNR to determine when a weight is no longer producing an informative gradient and only then to consider it for weight magnitude pruning. Like DPF, we compute a global magnitude based mask (MAG) based on the unpruned weights. The decision to prune is then taken if the SNR mask and MAG mask agree to prune a weight. This operation performs a logical AND and only prunes weights that are both converged as well as small. As the SNR is computed per neuron and gradually decreases below 1 during training of the weight, our method does not need a manual sparsity schedule. In particular, we set the sparsity of the MAG mask according to the sparsity arising automatically from the SNR mask, which allows us to prune less in the beginning and more at the end of training. At any stage of the pruning process, the algorithm has access to both gradient as well as weight magnitude information and is thus able to adapt to the dynamics of the optimization process (Algorithm 1).

To avoid overpruning towards the end of training when the SNR of most weights transitions from $> 1$ to $< 1$, we put the pruning on hold based on the training loss, which deactivates pruning when the training loss increases. Pruning is resumed, when the best so far seen training loss is reached again. Although this heuristic seems to work reasonably well for the datasets we tried, and effectively ties the total sparsity to the loss, learning a global sparsity parameter during dynamic pruning is so far unsolved, and we leave finding a more elegant solution to overpruning for future work.

As a limitation of our method we require more memory compared to DPF, since we do not only need to save the pruned and unpruned model like DPF, but in addition also the gradient SNR for each weight. Further, our computational requirements also increase by a multiplicative factor of approximately 1.25 (Dangel et al., 2020) since we compute the SNR per gradient computation.

## 5. Experiments

In our experiments, we directly compare to DPF (Lin et al., 2019) and normal dense training. In addition, we compare with GRASP-ABS, a variant of GRASP (Evci et al., 2020) proposed by Frankle

et al. (2021) which uses the absolute value of the score in order to make GRASP more reliable. In section 5.1, we test our pruning method on MNIST (LeCun et al., 2010), FashionMNIST (Xiao et al., 2017) and Cifar-10 (Krizhevsky et al., 2009) using a simple Convolutional Neural Network and ResNet18 (He et al., 2016). In section 5.2 we perform a similar analysis on tabular data from OpenML (Vanschoren et al., 2013) using multi-layer perceptrons. Details in Appendix A.1.

## 5.1 Image data

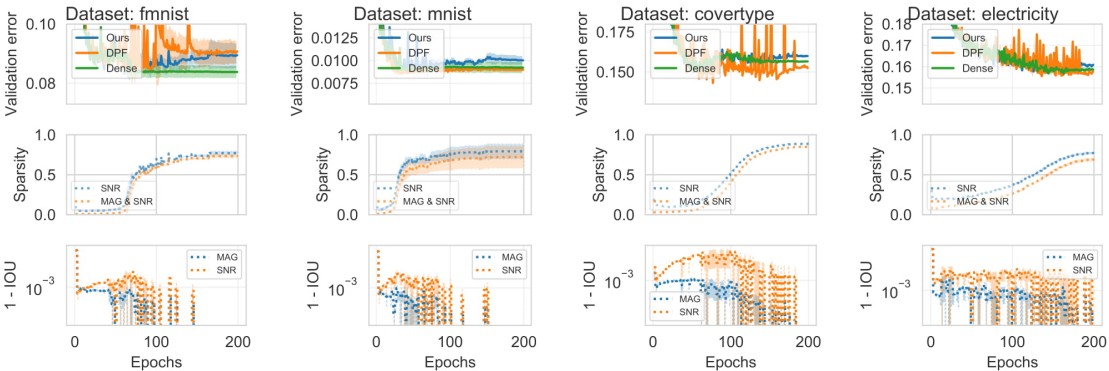

Figure 1: Results for FashionMNIST, MNIST, COVERTYPE and ELECTRICITY comparing our method to regular dense training and DPF for two different target sparsity ratios. Results averaged over 3 random seeds, DPF with target sparsity of 0.8. In the first row we plot the validation error at each epoch and we find that our method performs on par with DPF and normal dense training. The second row visualizes the global sparsity of the model at each epoch, which varies depending on the dataset because of the gradient SNR. The third row shows the IOU between the new and old mask at every mask update.

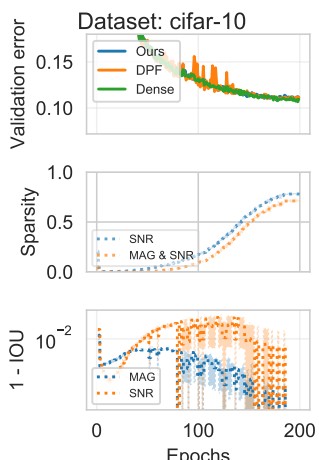

Figure 2: Results on CIFAR-10 dataset. Averaged over 3 random seeds, DPF with target sparsity of 0.8

The results on FashionMNIST and MNIST are shown on the left in Figure 1. Our algorithm compares favorably with DPF and dense training. Compared to DPF's schedule, our inferred schedule tends to prune many irrelevant weights at the beginning of training and the SNR mask may momentarily reduce the sparsity during training, which is not possible for DPF's mask. The initial pruning is followed by several consolidation epochs, in which the SNR mask will only slowly increase in sparsity. This consolidation period is shorter for MNIST compared to FashionMNIST, pointing to the intuition, that MNIST is easier to solve than FashionMNIST, which in turn allows to remove parameters more quickly.

An ablation experiment using only the magnitude mask or only the SNR mask for pruning is shown in Figure 3 in the Appendix. Pruning based only on the SNR mask leads to eventual overpruning and pruning based only on the MAG mask with a fixed sparsity prunes too many weights too early.

We also visualize the intersection over union of consecutive masks in the lower plots of each Figure measured in terms of their intersection over union (IOU) before and after a mask update. We show 1 - IOU in order to use a log scale as the mask changes happen at different magnitudes during training. Similarly to (Frankle et al., 2019b), we found that the weight magnitude mask (MAG) changes less towards the end of training. In contrast, the SNR mask changes

more towards the end of training, which is to be expected as more weights will converge and hence their SNR will be less than 1.

The results for pruning a ResNet-18 on CIFAR-10 are shown in Figure 2. Our method has the same final performance as DPF, but training is less noisy and at a similar speed as the dense model.

## 5.2 Tabular data

The results obtained on Electricity and Covertype from OpenML Vanschoren et al. (2013) are shown in the right half of Figure 1. DPF performs better than our method and dense training for Covertype and on par to our method and dense on Electricity.

## 5.3 A different weight update strategy

In the course of our experiments, we also implemented a different strategy for weight updates which differs from DPF (Lin et al., 2019). For context: In DPF a weight $\boldsymbol{\theta}_i$ may become unpruned if one of the following conditions holds:

1. If a different weight $\boldsymbol{\theta}_j$ decreases in magnitude, below the current cutoff threshold, and the weight $\boldsymbol{\theta}_i$ is close to this threshold then the weight is unpruned.

2. In DPF, gradient updates are also performed for pruned weights, hence a weight may become unpruned if the gradient update pushes its magnitude back into the current magnitude mask.

In our experiments, we investigated our proposed method and DPF if we only use condition (1), which hence reduces the chance of unpruning a weight during training. The results are shown in Figures 5 and 6 in the Appendix. DPF is noticeably less noisy in this setting, otherwise the final performances for our method and DPF are almost identical, apart from CIFAR-10 where DPF works worse in this setting. For CIFAR-10, our method also reaches higher final sparsities than in the results shown in Section 5.1. Since the SNR criterion ensures that only small *and* converged weights, with gradients resembling white noise, are being pruned, we conjecture that updating them still after being pruned mostly adds disruptive noise into the network, hence not updating them (dropping condition 2.) smoothes the curves without loosing performance. DFP in contrast may suffer from this change, since weight magnitude pruning alone may prune weights that did not reach their final value yet and hence updating them (at the expense of a noisy learning curve) is more crucial.

## 6. Conclusion & Future Work

We proposed the signal-to-noise ratio of mini-batch gradients as a new pruning criterion, which measures when a weight is no longer producing an informative gradient. We explain how to extend DPF (Lin et al., 2019) using the new criterion which allows the network to decrease its sparsity if necessary, which makes the pruning more flexible. In our experiments, we demonstrate that we can obtain results on par with normal dense training and DPF on image and tabular data, while exposing no free parameter. We leave as future work the question on how to compute the SNR for more complex networks and how to prevent overpruning more elegantly than using early stopping on the pruning masks.

## 7. Broader impact

We see opportunities of our work for positive impact, by contributing to potential energy savings in training neural networks and making them more memory efficient, hence reducing their carbon footprint and financial cost. However, our work may also lead to negative side effects. For example,

while our method produces sparse networks, the resulting classifiers may not necessarily be fair and could for example give dominating groups in a population even greater weight, than non-sparse networks.

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

# Appendix A. Appendix

## A.1 Experimental details

### A.1.1 IMAGE DATA

All experiments for MNIST and FashionMNIST use a small CNN with 4 convolutional layers with relu activation function, each followed by max pooling, and a final linear layer. The number of filters doubles between consecutive filters with an initial filter count of 16.

An initial learning rate of 1e-2 which decays according to a cosine schedule (Loshchilov and Hutter, 2016) to 1e-4 and is used with the Adam optimizer (Kingma and Ba). An additional L2 regularization of 1e-4 is used for all experiments.

## A.2 Tabular data

We use multi-layer perceptrons with 7 layers and a width of 512 to be sufficiently overparameterized. The optimization was carried out with the same parameters as for image data, however with an initial learning rate of 1e-1.

## A.3 Ablation study

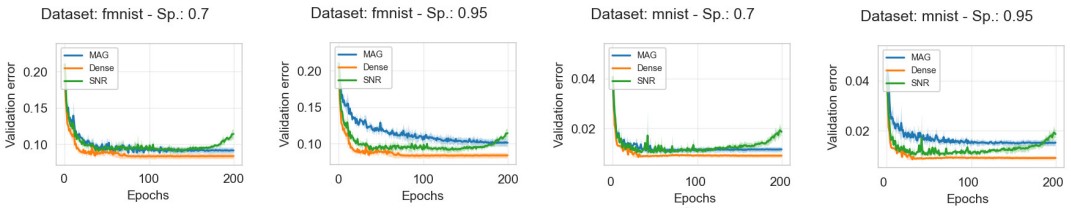

Figure 3: Ablation experiment when only using a magnitude mask or the dynamic SNR mask for pruning. To illustrate the behavior no early stopping on the pruning level is performed.

## A.4 Mask update strategies

DPF (Lin et al., 2019) used an extra hyperparameter search to determine the number of SGD steps between mask updates to be 16. However, depending on the dataset and network architecture the default may no longer be optimal. Therefore, we investigate the impact of the frequency of mask updates on the optimization. We compare dense training to updating the mask after every and after 32 and 128 SGD steps. Note, that in these experiments we do not perform any early stopping on the pruning level.

We find that the number of mask updates has only a small influence on the optimization overall and only slightly changes the maximum attained sparsity level. More mask updates tend to lead to higher levels of sparsity in particular for the openml datasets electricity and covertype for which we use MLPs. For the ConvNets used for MNIST and FMNIST .

## A.5 Ablation on weight update strategy

The effect of the alternative weight update strategy as outlined in Section 5.3 in the main paper is shown in Figure 5 for CIFAR-10, and in Figure 6 for MNIST, FashionMNIST, COVERTYPE and ELECTRICITY.

## A.6 Effect of the number of burn-in epochs and the memory width of the exp. mov. avg of the gradient SNR

The gradient SNR can be a noisy quantity. Hence, in the algorithm we smooth it with an exponential moving average with a fixed factor $\gamma \in [0, 1]$ over optimization steps as proposed by Mahsereci et al. (2017). In this section, we investigate what influence the parameter $\gamma$ has on the pruning algorithm, and motivate a non-sensitive default. The smoothed SNR is computed as follows: $snr_t^{smooth} = \gamma snr_{t-1}^{smooth} + (1 - \gamma)snr_t(\boldsymbol{\theta})$. The smoothing factor $\gamma$ has a simple interpretation. Roughly after $m$ steps, the contribution of the SNR value $snr_{t-m}$ of previous iteration $t - m$ will have decayed to about $\gamma^m$ percent relative to the current value $snr_t$. This number of iterations $m$ is roughly the 'memory' of the smoother. We want the memory to be as large as possible (to reduce as much noise as possible) much small enough such that the smoothed SNR is flexible, that is the memory should be small in comparison to the total length $T$ of the optimization run. The formula is thus $\gamma = (0.01)^{-m}$ for a residual contribution of 1%. The relative memory $r_m := \frac{m}{T}$ can be set to a meaningful default. The smaller $r_m$, the smaller $\gamma$ resulting in less smoothing.

A second parameter is the number of burn-in steps $bin_{steps}$ after which we start pruning. The results of the ablation are shown in Figure 7 for varying $r_m \in \{0.02, 0.1\}$ which equals 1/50-th and 1/10-th of the total run respectively, and $bin_{steps} \in \{1, 10\}$. We find that a large window of $r_m = 1/10$ prunes too cautiously, and the smaller window $r_m = 1/50$ works well across all tested datasets and networks. The number of burning steps $bin_{steps}$ seems to have no effect on the final pruned network. Hence, as defaults for all experiments we chose $r_m = 0.02 = 1/50$ and $bin_{steps} = 1$ epoch.

## A.7 Layer-wise pruning ratios of our method

In Figures 8 and 9 we visualize the sparsity per layer at each training epoch for our proposed method. We find that the SNR mask tends to want to prune more layers in the layers closer to the input and less closer to the output, while the reverse is true for the MAG mask.

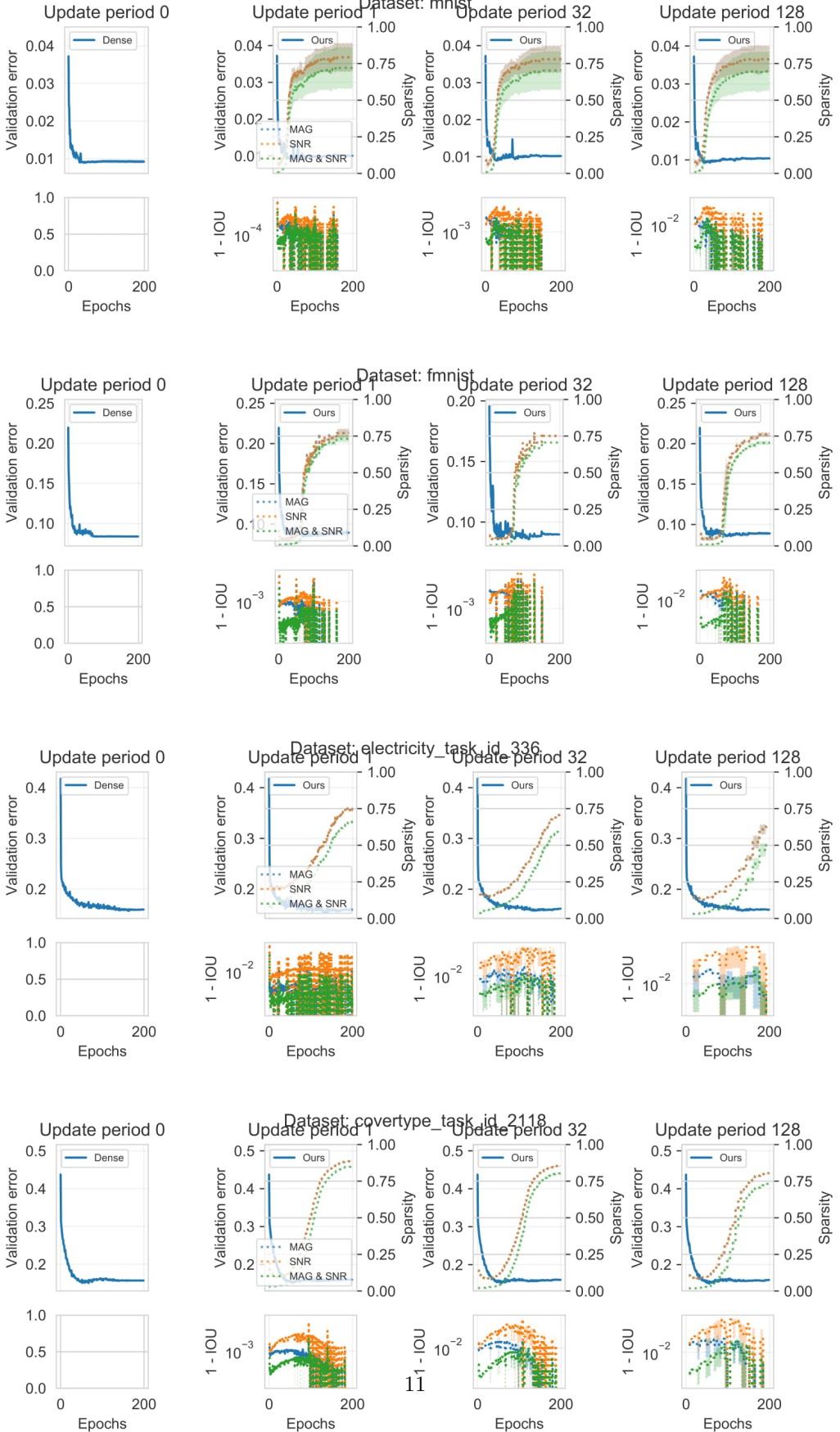

Figure 4: Ablation study on the influence of the number of SGD steps between mask updates.

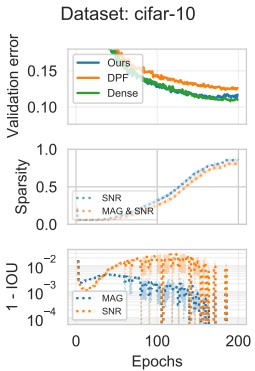

Figure 5: Results on CIFAR-10 dataset. Averaged over 3 random seeds, DPF with target sparsity of 0.8

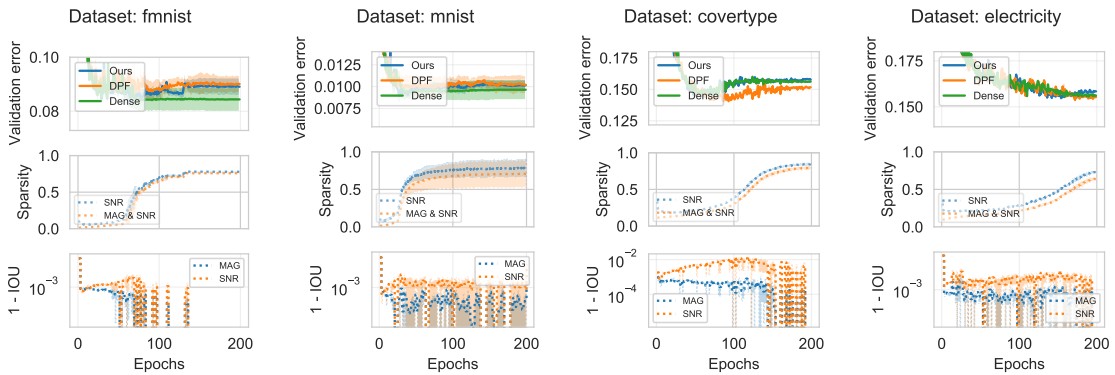

Figure 6: Results for FashionMNIST and MNIST comparing our method to regular dense training and DPF for two different target sparsity ratios. Results averaged over 3 random seeds, DPF with target sparsity of 0.8

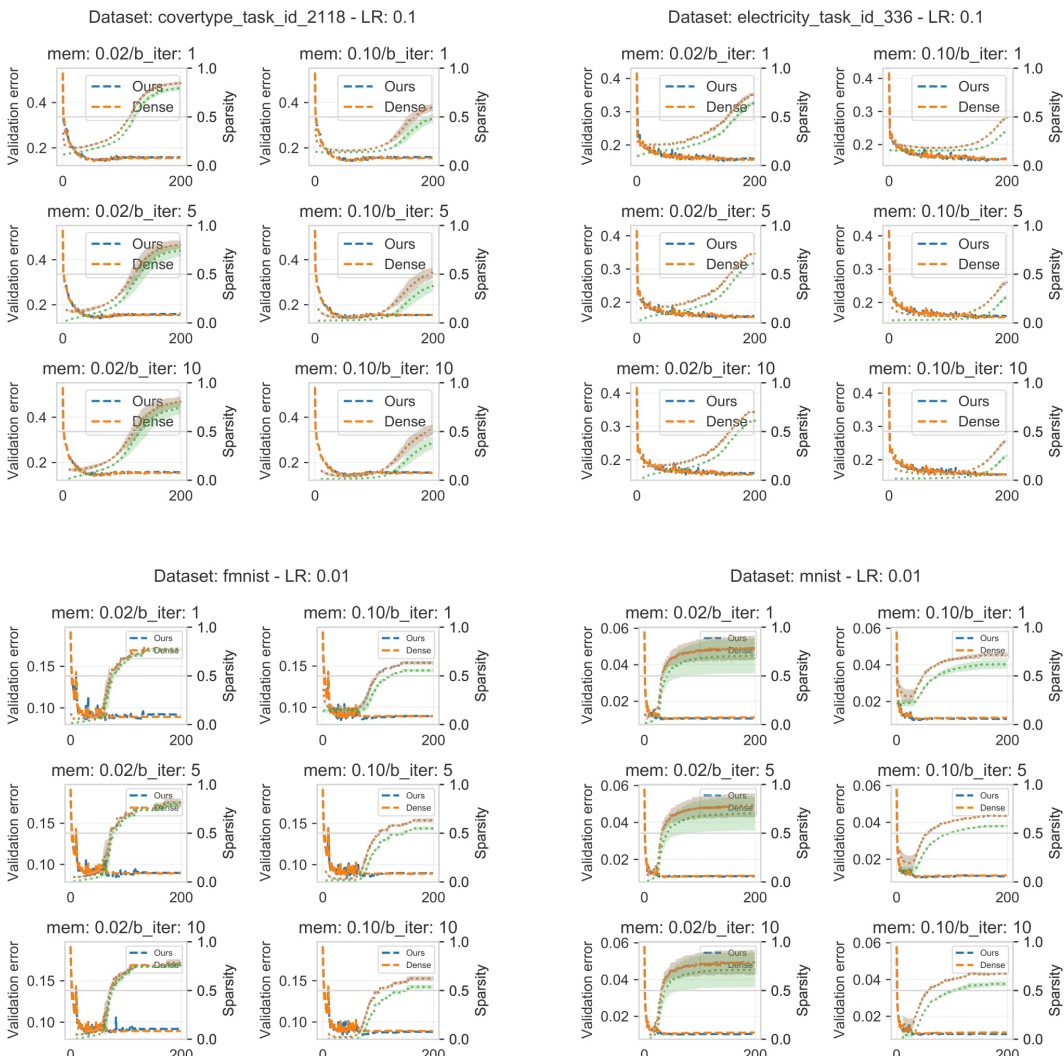

Figure 7: Ablation study on the influence of the burn in steps $bin_{steps}$ and the memory fraction $r_m$ of the exp. mov. avg of the SNR.

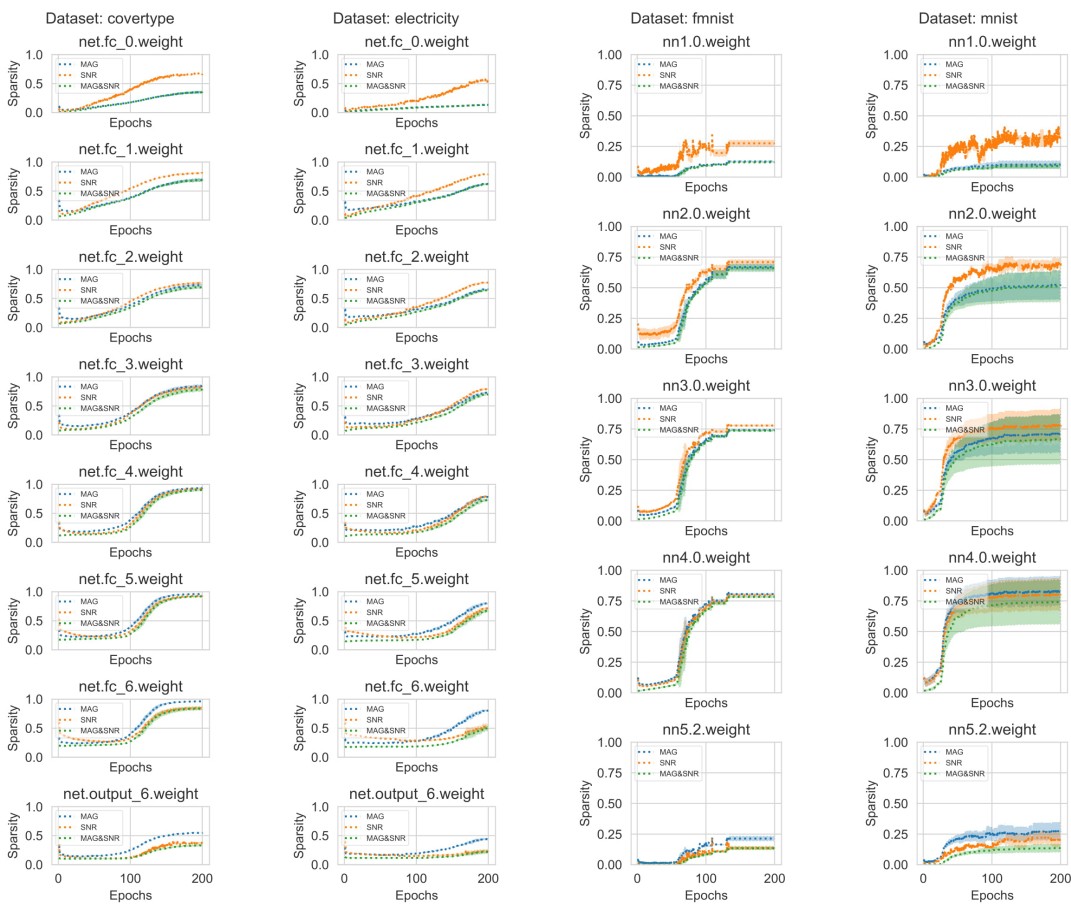

Figure 8: Sparsities per layer found by our method during training for the magnitude (MAG), the mini-batch gradient SNR (SNR) and the combined mask (MAG& SNR).

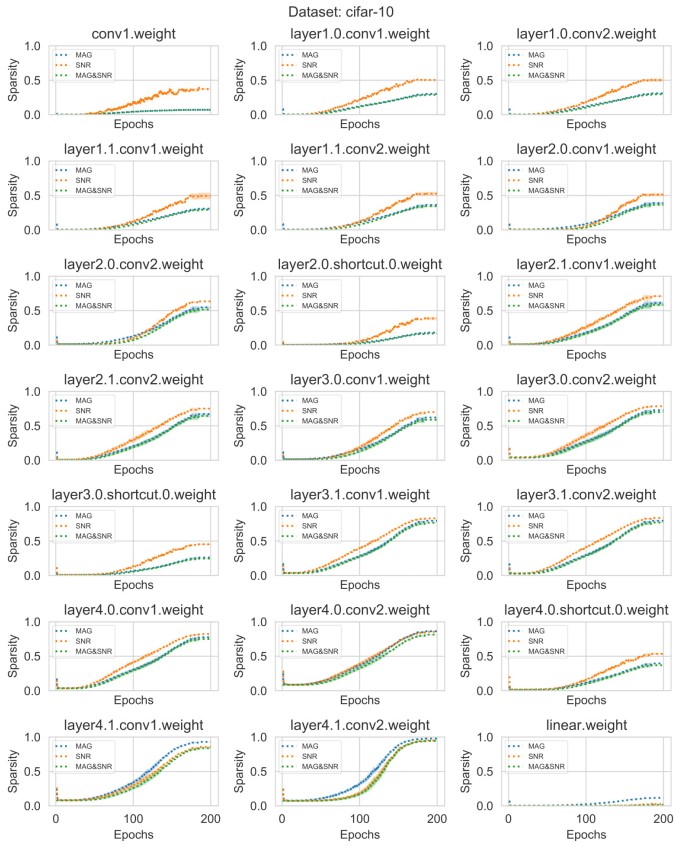

Figure 9: Sparsities per layer found by our method for CIFAR-10.

