# OpenReview forum: "Dynamic Pruning of a Neural Network via Gradient Signal-to-Noise Ratio"
_ICML.cc/2021/Workshop/AutoML — AutoML@ICML2021 Poster_

### Official Review · Reviewer_Q1uD · 2021-06-09
**Core idea easy to understand, not enough information to be reproducible.**

**Rating:** 5
**Confidence:** 5

**Review:**

The core idea is to estimate the information content of the gradients. Only weights that receive no meaningful updates and are small can be pruned.
I think the idea is conceptually interesting and the results are fine. I would not be able to reproduce any of the results due to lack of information. I think the biggest issue with the paper is how it is written. Many aspects are not explained in sufficient detail.
If the issues highlighted in this review get addressed prior to the final version, I think it is ok as a work in progress workshop submission, but for a full contribution all the mistakes and confusing aspects need to be resolved.


It is unclear to me how the snr is computed. I have the impression that the snr is the square of the gradient (averaged over the batch) is simply divided by the average of the squared (and possibly centered) gradients in the batch. In essence in the numerate squaring follows averaging, in the denumerator averaging follows squaring.

The paper states that : If the statistic z(θ) := log p(∇LB(θ)|∇L(θ) = 0) coincides with its expectation Ex[z(θ)], ....
How is  Ex[z(θ)] computed? What does it mean to coincide in this case. My interpretation would be that the gradients have the same maginitude as random noise, but I have the impression that the authors mean that coincides means that the gradients are not larger than what is expected. It would be helpful to define this propperly.

All this estimation depends on the batch size and can of course also interact with batch normalization etc etc. However, at no point the batch size is actually provided.

I was not able to find what the magnitude cutoff is or how this parameter is set/optimized.

Algorithm 1: line 5, the authors use the or operator, the description has the and operator.

Algorithm 1: lines 3,4 The m_snr <- m_snr,t(\theta_t)  and the equivalent on the other line are not helpful at all.
There is no explicit description of m_snr,t(\theta_t) in the paper.

Section 5.3 talks about results in the appendix. I would remove this section and ensure that all the other detail is actually explained properly.

The broader impact section states that the approach can contribute to potential energy savings in training neural networks. However, the paper also states that training became more expensive. I assume the authors meant to say deploying instead of training.

---

### Official Review · Reviewer_wzSB · 2021-06-17
**Review for Dynamic Pruning of a Neural Network via Gradient Signal-to-Noise Ratio**

**Rating:** 6
**Confidence:** 3

**Review:**

+: This work proposes a new sparsity criterion to allow the adaptive mask during the training. It performs
weight magnitude pruning according to a learnable pruning schedule during training, allowing less pruning at the beginning and more pruning towards the end. The author[s] proposes a heuristics based on the best so far training loss to decide when to stop and resume the pruning.  Results on small image datasets and some datasets in OpenML show its effectiveness.


-: The reason for using training loss for early-stopping and resume pruning is not well justified, e.g. when is the best time to observe the training loss? if step-wise is a better metric for SNR estimation compared to epoch-wise? How it behaves for efficient models such as MobileNet-v3?

---

### Official Review · Reviewer_b1DF · 2021-06-17
**An improved dynamic pruning method that performs well on a suite of different tasks.**

**Rating:** 7
**Confidence:** 3

**Review:**

Clarity
Overall the text is clear and the method is easy to understand from the writing. The authors also have a good related work and do a good job of contextualizing their method in the literature. The writing is still rough in certain places with non-grammatical text and typos.


Pros / Originality / Significance
This method works on a method in a promising sparsity space: one where pruned weights can be dynamically added back in over the course of training.
Author's proposed method get rid of a real drawback of the Dynamic Pruning with Feedback, where the sparsity schedule over training has to be manually set and is quite sensitive to hyperparmeter values.
Their method also takes into account gradients into account when pruning weights, as they contain more information than randomly initialized weights at the start of training. This is a nice principled additional to the prior sparsity work they build off of.
The authors conduct extensive experiments to try their new pruning method on a wide variety of different datasets. Their proposed method does better than prior state-of-the-art pruning methods on a convincing suite of downstream tasks, while having less hyperparmeters.


Cons
The authors state their method requires more memory compared to DPF along with more compute. It would be nice to quantitatively show these how much more it is in Table 1 for example.
The method does rely on some heuristics like early stopping to avoid overpruning as their method is tied to the training loss. It would be nice to validate this method on different models to make sure these heuristics hold for a more varied setup

---

### Decision · Program_Chairs · 2021-06-21

Accept (Poster)